# Reliable Decision Support using Counterfactual Models

**Peter Schulam**
Department of Computer Science
Johns Hopkins University
Baltimore, MD 21211
pschulam@cs.jhu.edu

**Suchi Saria**
Department of Computer Science
Johns Hopkins University
Baltimore, MD 21211
ssaria@cs.jhu.edu

## Abstract

Decision-makers are faced with the challenge of estimating what is likely to happen when they take an action. For instance, if I choose not to treat this patient, are they likely to die? Practitioners commonly use supervised learning algorithms to fit predictive models that help decision-makers reason about likely future outcomes, but we show that this approach is unreliable, and sometimes even dangerous. The key issue is that supervised learning algorithms are highly sensitive to the policy used to choose actions in the training data, which causes the model to capture relationships that do not generalize. We propose using a different learning objective that predicts *counterfactuals* instead of predicting outcomes under an existing action policy as in supervised learning. To support decision-making in temporal settings, we introduce the Counterfactual Gaussian Process (CGP) to predict the counterfactual future progression of continuous-time trajectories under sequences of future actions. We demonstrate the benefits of the CGP on two important decision-support tasks: risk prediction and "what if?" reasoning for individualized treatment planning.

## 1 Introduction

Decision-makers are faced with the challenge of estimating what is likely to happen when they take an action. One use of such an estimate is to evaluate *risk*; e.g. is this patient likely to die if I do not intervene? Another use is to perform "what if?" reasoning by comparing outcomes under alternative actions; e.g. would changing the color or text of an ad lead to more click-throughs? Practitioners commonly use supervised learning algorithms to help decision-makers answer such questions, but these decision-support tools are unreliable, and can even be dangerous.

Consider, for instance, the finding discussed by Caruana et al. [2015] regarding risk of death among those who develop pneumonia. Their goal was to build a model that predicts risk of death for a hospitalized individual with pneumonia so that those at high-risk could be treated and those at low-risk could be safely sent home. Their model counterintuitively learned that asthmatics are less likely to die from pneumonia. They traced the result back to an *existing policy* that asthmatics with pneumonia should be directly admitted to the intensive care unit (ICU), therefore receiving more aggressive treatment. Had this model been deployed to assess risk, then asthmatics might have received *less* care, putting them at greater risk. Caruana et al. [2015] show how these counterintuitive relationships can be problematic and ought to be addressed by "repairing" the model. We note, however, that these issues stem from a deeper limitation: when training data is affected by actions, supervised learning algorithms capture relationships caused by action policies, and these relationships do not generalize when the policy changes.

To build reliable models for decision support, we propose using learning objectives that predict *counterfactuals*, which are collections of random variables $\{Y[a] : a \in \mathcal{C}\}$ used in the *potential*

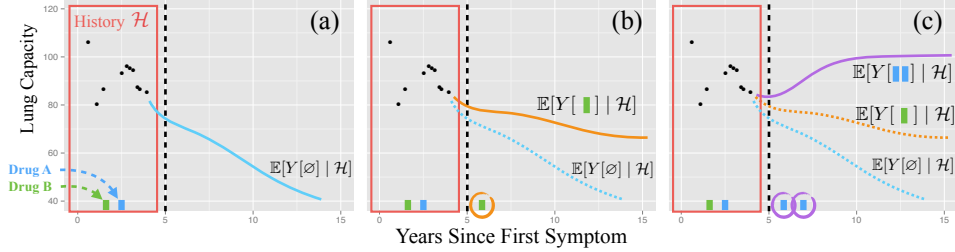

Figure 1: Best viewed in color. An illustration of the counterfactual GP applied to health care. The red box in (a) shows previous lung capacity measurements (black dots) and treatments (the history). Panels (a)-(c) show the type of predictions we would like to make. We use $Y[a]$ to represent the potential outcome under action $a$.

*outcomes* framework [Neyman, 1923, 1990, Rubin, 1978]. Counterfactuals model the outcome $Y$ after an action $a$ is taken from a set of choices $\mathcal{C}$. Counterfactual predictions are broadly applicable to a number of decision-support tasks. In medicine, for instance, when evaluating a patient's risk of death $Y$ to determine whether they should be treated aggressively, we want an estimate of how they will fare *without* treatment. This can be done by predicting the counterfactual $Y[\varnothing]$, where $\varnothing$ stands for "do nothing". In online marketing, to decide whether we should display ad $a_1$ or $a_2$, we may want an estimate of click-through $Y$ under each, which amounts to predicting $Y[a_1]$ and $Y[a_2]$.

To support decision-making in temporal settings, we develop the Counterfactual Gaussian Process (CGP) to predict the counterfactual future progression of continuous-time trajectories under sequences of future actions. The CGP can be learned from and applied to time series data where actions are taken and outcomes are measured at irregular time points; a generalization of discrete time series. Figure 1 illustrates an application of the CGP. We show an individual with a lung disease, and would like to predict her future lung capacity (y-axis). Panel (a) shows the *history* in the red box, which includes previous lung capacity measurements (black dots) and previous treatments (green and blue bars). The blue counterfactual trajectory shows what might occur under *no action*, which can be used to evaluate this individual's risk. In panel (b), we show the counterfactual trajectory under a single future green treatment. Panel (c) illustrates "what if?" reasoning by overlaying counterfactual trajectories under two different action sequences; in this case it seems that two future doses of the blue drug may lead to a better outcome than a single dose of green.

**Contributions.** Our key methodological contribution is the Counterfactual Gaussian process (CGP), a model that predicts how a continuous-time trajectory will progress under sequences of actions. We derive an adjusted maximum likelihood objective that learns the CGP from *observational traces*; irregularly sampled sequences of actions and outcomes denoted using $\mathcal{D} = \{\{(y_{ij}, a_{ij}, t_{ij})\}_{j=1}^{n_i}\}_{i=1}^{m}$, where $y_{ij} \in \mathbb{R} \cup \{\varnothing\}$, $a_{ij} \in \mathcal{C} \cup \{\varnothing\}$, and $t_{ij} \in [0, \tau]$.[1] Our objective accounts for and removes the effects of the policy used to choose actions in the observational traces. We derive the objective by jointly modeling observed actions and outcomes using a marked point process (MPP; see e.g., Daley and Vere-Jones 2007), and show how it correctly learns the CGP under a set of assumptions analogous to those required to learn counterfactual models in other settings.

We demonstrate the CGP on two decision-support tasks. First, we show how the CGP can make reliable risk predictions that do not depend on the action policy in the training data. On the other hand, we show that predictions made by models trained using classical supervised learning objectives are sensitive to the policies. In our second experiment, we use data from a real intensive care unit (ICU) to learn the CGP, and qualitatively demonstrate how the CGP can be used to compare counterfactuals and answer "what if?" questions, which could offer medical decision-makers a powerful new tool for individualized treatment planning.

## 1.1 Related Work

Decision support is a rich field; because our main methodological contribution is a counterfactual model for time series data, we limit the scope of our discussion of related work to this area.

**Causal inference.** Counterfactual models stem from causal inference. In that literature, the difference between the counterfactual outcomes if an action had been taken and if it had not been taken

is defined as the *causal effect* of the action (see e.g., Pearl 2009 or Morgan and Winship 2014). *Potential outcomes* are commonly used to formalize counterfactuals and obtain causal effect estimates [Neyman, 1923, 1990, Rubin, 1978]. Potential outcomes are often applied to cross-sectional data; see, for instance, the examples in Morgan and Winship 2014. Recent examples from the machine learning literature are Bottou et al. [2013] and Johansson et al. [2016].

**Potential outcomes in discrete time.** Potential outcomes have also been used to estimate the causal effect of a sequence of actions in discrete time on a final outcome (e.g. Robins 1986, Robins and Hernán 2009, Taubman et al. 2009). The key challenge in the sequential setting is to account for feedback between intermediate outcomes that determine future treatment. Conversely, Brodersen et al. [2015] estimate the effect that a *single discrete intervention* has on a *discrete* time series. Recent work on optimal dynamic treatment regimes uses the sequential potential outcomes framework proposed by Robins [1986] to learn lists of discrete-time treatment rules that optimize a scalar outcome. Algorithms for learning these rules often use action-value functions (Q-learning; e.g., Nahum-Shani et al. 2012). Alternatively, A-learning is a semiparametric approach that directly learns the relative difference in value between alternative actions [Murphy, 2003].

**Potential outcomes in continuous time.** Others have extended the potential outcomes framework in Robins [1986] to learn causal effects of actions taken in continuous-time on a single final outcome using observational data. Lok [2008] proposes an estimator based on structural nested models [Robins, 1992] that learns the instantaneous effect of administering a single type of treatment. Arjas and Parner [2004] develop an alternative framework for causal inference using Bayesian posterior predictive distributions to estimate the effects of actions in continuous time on a final outcome. Both Lok [2008] and Arjas and Parner [2004] use marked point processes to formalize assumptions that make it possible to learn causal effects from continuous-time observational data. We build on these ideas to learn causal effects of actions on continuous-time *trajectories* instead of a single outcome. There has also been recent work on building expressive models of treatment effects in continuous time. Xu et al. [2016] propose a Bayesian nonparametric approach to estimating individual-specific treatment effects of discrete but irregularly spaced actions, and Soleimani et al. [2017] model the effects of continuous-time, continuous-valued actions. Causal effects in continuous-time have also been studied using differential equations. Mooij et al. [2013] formalize an analog of Pearl's "do" operation for deterministic ordinary differential equations. Sokol and Hansen [2014] make similar contributions for stochastic differential equations by studying limits of discrete-time non-parametric structural equation models [Pearl, 2009]. Cunningham et al. [2012] introduce the Causal Gaussian Process, but their use of the term "causal" is different from ours, and refers to a constraint that holds for sample paths of the GP.

**Reinforcement learning.** Reinforcement learning (RL) algorithms learn from data where actions and observations are interleaved in discrete time (see e.g., Sutton and Barto 1998). In RL, however, the focus is on learning a *policy* (a map from states to actions) that optimizes the expected reward, rather than a model that predicts the effects of the agent's actions on future observations. In model-based RL, a model of an action's effect on the subsequent state is produced as a by-product either offline before optimizing the policy (e.g., Ng et al. 2006) or incrementally as the agent interacts with its environment. In most RL problems, however, learning algorithms rely on active experimentation to collect samples. This is not always possible; for example, in healthcare we cannot actively experiment on patients, and so we must rely on retrospective observational data. In RL, a related problem known as off-policy evaluation also uses retrospective observational data (see e.g., Dudík et al. 2011, Swaminathan and Joachims 2015, Jiang and Li 2016, Păduraru et al. 2012, Doroudi et al. 2017). The goal is to use state-action-reward sequences generated by an agent operating under an unknown policy to estimate the expected reward of a target policy. Off-policy algorithms typically use action-value function approximation, importance reweighting, or doubly robust combinations of the two to estimate the expected reward.

## 2 Counterfactual Models from Observational Traces

Counterfactual GPs build on ideas from potential outcomes [Neyman, 1923, 1990, Rubin, 1978], Gaussian processes [Rasmussen and Williams, 2006], and marked point processes [Daley and Vere-Jones, 2007]. In the interest of space, we review potential outcomes and marked point processes, but refer the reader to Rasmussen and Williams [2006] for background on GPs.

**Background: Potential Outcomes.** To formalize counterfactuals, we adopt the potential outcomes framework [Neyman, 1923, 1990, Rubin, 1978], which uses a collection of random variables $\{Y[a] :$

$a \in \mathcal{C}\}$ to model the outcome after each action $a$ from a set of choices $\mathcal{C}$. To make counterfactual predictions, we must learn the distribution $P(Y[a] \mid X)$ for each action $a \in \mathcal{C}$ given features $X$. If we can freely experiment by repeatedly taking actions and recording the effects, then it is straightforward to fit a predictive model. Conducting experiments, however, may not be possible. Alternatively, we can use observational data, where we have example actions $A$, outcomes $Y$, and features $X$, but do not know how actions were chosen. Note the difference between the action $a$ and the random variable $A$ that models the *observed actions* in our data; the notation $Y[a]$ serves to distinguish between the observed distribution $P(Y \mid A, X)$ and the target distribution $P(Y[a] \mid X)$.

In general, we can only use observational data to estimate $P(Y \mid A, X)$. Under two assumptions, however, we can show that this conditional distribution is equivalent to the counterfactual model $P(Y[a] \mid X)$. The first is known as the Consistency Assumption.

**Assumption 1** (Consistency). *Let $Y$ be the observed outcome, $A \in \mathcal{C}$ be the observed action, and $Y[a]$ be the potential outcome for action $a \in \mathcal{C}$, then:* $(Y \triangleq Y[a]) \mid A = a$.

Under consistency, we have that $P(Y \mid A = a) = P(Y[a] \mid A = a)$. Now, the potential outcome $Y[a]$ may depend on the action $A$, so in general $P(Y[a] \mid A = a) \neq P(Y[a])$. The next assumption posits that the features $X$ include all possible *confounders* [Morgan and Winship, 2014], which are sufficient to d-separate $Y[a]$ and $A$.

**Assumption 2** (No Unmeasured Confounders (NUC)). *Let $Y$ be the observed outcome, $A \in \mathcal{C}$ be the observed action, $X$ be a vector containing all potential confounders, and $Y[a]$ be the potential outcome under action $a \in \mathcal{C}$, then:* $(Y[a] \perp A) \mid X$.

Under Assumptions 1 and 2, $P(Y \mid A, X) = P(Y[a] \mid X)$. An extension of Assumption 2 introduced by Robins [1997] known as *sequential NUC* allows us to estimate the effect of a sequence of actions in discrete time on a single outcome. In continuous-time settings, where both the type and *timing* of actions may be statistically dependent on the potential outcomes, Assumption 2 (and sequential NUC) cannot be applied as-is. We will describe an alternative that serves a similar role for CGPs.

**Background: Marked Point Processes.** Point processes are distributions over sequences of times-tamps $\{T_i\}_{i=1}^N$, which we call points, and a marked point process (MPP) is a point process where each point is annotated with an additional random variable $X_i$, called its mark. For example, a point $T$ might represent the arrival time of a customer, and $X$ the amount that she spent at the store. We emphasize that both the annotated points $(T_i, X_i)$ and the number of points $N$ are random variables.

A point process can be characterized as a counting process $\{N_t : t \geq 0\}$ that counts the number of points that occured up to and including time $t$: $N_t = \sum_{i=1}^N \mathbb{I}_{(T_i \leq t)}$. By definition, this processes can only take integer values, and $N_t \geq N_s$ if $t \geq s$. In addition, it is commonly assumed that $N_0 = 0$ and that $\Delta N_t = \lim_{\delta \to 0^+} N_t - N_{t-\delta} \in \{0, 1\}$. We can parameterize a point process using a probabilistic model of $\Delta N_t$ given the history of the process $\mathcal{H}_{t^-}$ up to but not including time $t$ (we use $t^-$ to denote the left limit of $t$). Using the Doob-Meyer decomposition [Daley and Vere-Jones, 2007], we can write $\Delta N_t = \Delta M_t + \Delta \Lambda_t$, where $M_t$ is a martingale, $\Lambda_t$ is a cumulative intensity function, and

$$P(\Delta N_t = 1 \mid \mathcal{H}_{t^-}) = \mathbb{E}[\Delta N_t \mid \mathcal{H}_{t^-}] = \mathbb{E}[\Delta M_t \mid \mathcal{H}_{t^-}] + \Delta \Lambda_t(\mathcal{H}_{t^-}) = 0 + \Delta \Lambda_t(\mathcal{H}_{t^-}),$$

which shows that we can parameterize the point process using the conditional intensity function $\lambda^*(t) \, \mathrm{d}t \triangleq \Delta \Lambda_t(\mathcal{H}_{t^-})$. The star superscript on the intensity function serves as a reminder that it depends on the history $\mathcal{H}_{t^-}$. For example, in non-homogeneous Poisson processes $\lambda^*(t)$ is a function of time that does not depend on the history. On the other hand, a Hawkes process is an example of a point process where $\lambda^*(t)$ *does* depend on the history [Hawkes, 1971]. MPPs are defined by an intensity that is a function of both the time $t$ and the mark $x$: $\lambda^*(t, x) = \lambda^*(t)p^*(x \mid t)$. We have written the joint intensity in a factored form, where $\lambda^*(t)$ is the intensity of *any* point occuring (that is, the mark is unspecified), and $p^*(x \mid t)$ is the pdf of the observed mark given the point's time. For an MPP, the history $\mathcal{H}_t$ contains each prior point's time and mark.

## 2.1 Counterfactual Gaussian Processes

Let $\{Y_t : t \in [0, \tau]\}$ denote a continuous-time stochastic process, where $Y_t \in \mathbb{R}$, and $[0, \tau]$ defines the interval over which the process is defined. We will assume that the process is observed at a discrete set of irregular and random times $\{(y_j, t_j)\}_{j=1}^n$. We use $\mathcal{C}$ to denote the set of possible *action types*, $a \in \mathcal{C}$ to denote the elements of the set, and define an action to be a 2-tuple $(a, t)$ specifying

an action type $a \in \mathcal{C}$ and a time $t \in [0, \tau]$ at which it is taken. To refer to multiple actions, we use $\mathbf{a} = [(a_1, t_1), \ldots, (a_n, t_n)]$. Finally, we define the history $\mathcal{H}_t$ at a time $t \in [0, \tau]$ to be a list of all previous observations of the process and all previous actions. Our goal is to model the counterfactual:

$$P(\{Y_s[\mathbf{a}] : s > t\} \mid \mathcal{H}_t), \text{ where } \mathbf{a} = \{(a_j, t_j) : t_j > t\}_{j=1}^m. \tag{1}$$

To learn the counterfactual model, we will use *traces* $\mathcal{D} \triangleq \{\mathbf{h}_i = \{(t_{ij}, y_{ij}, a_{ij})\}_{j=1}^{n_i}\}_{i=1}^m$, where $y_{ij} \in \mathbb{R} \cup \{\varnothing\}$, $a_{ij} \in \mathcal{C} \cup \{\varnothing\}$, and $t_{ij} \in [0, \tau]$. Our approach is to model $\mathcal{D}$ using a marked point process (MPP), which we learn using the traces. Using Assumption 1 and two additional assumptions defined below, the estimated MPP recovers the counterfactual model in Equation 1.

We define the MPP mark space as the Cartesian product of the outcome space $\mathbb{R}$ and the set of action types $\mathcal{C}$. To allow either the outcome or the action (but not both) to be the null variable $\varnothing$, we introduce binary random variables $z_y \in \{0, 1\}$ and $z_a \in \{0, 1\}$ to indicate when the outcome $y$ and action $a$ are not $\varnothing$. Formally, the mark space is $\mathcal{X} = (\mathbb{R} \cup \{\varnothing\}) \times (\mathcal{C} \cup \{\varnothing\}) \times \{0, 1\} \times \{0, 1\}$. We can then write the MPP intensity as

$$\lambda^*(t, y, a, z_y, z_a) = \underbrace{\lambda^*(t)p^*(z_y, z_a \mid t)}_{\text{[A] Event model}} \underbrace{p^*(y \mid t, z_y)}_{\text{[B] Outcome model (GP)}} \underbrace{p^*(a \mid y, t, z_a)}_{\text{[C] Action model}}, \tag{2}$$

where we have again used the $*$ superscript as a reminder that the hazard function and densities above are implicitly conditioned on the history $\mathcal{H}_{t^-}$. The parameterization of the event and action models can be chosen to reflect domain knowledge about how the timing of events and choice of action depend on the history. The outcome model is parameterized using a GP (or any elaboration such as a hierarchical GP or mixture of GPs), and can be treated as a standard regression model that predicts how the future trajectory will progress given the previous actions and outcome observations.

**Learning.** To learn the CGP, we maximize the likelihood of observational traces over a fixed interval $[0, \tau]$. Let $\boldsymbol{\theta}$ denote the model parameters, then the likelihood for a single trace is

$$\ell(\boldsymbol{\theta}) = \sum_{j=1}^n \log p_{\boldsymbol{\theta}}^*(y_j \mid t_j, z_{yj}) + \sum_{j=1}^n \log \lambda_{\boldsymbol{\theta}}^*(t_j) p_{\boldsymbol{\theta}}^*(a_j, z_{yj}, z_{aj} \mid t_j, y_j) - \int_0^\tau \lambda_{\boldsymbol{\theta}}^*(s) \, \mathrm{d}s. \tag{3}$$

We assume that traces are independent, and so can learn from multiple traces by maximizing the sum of the individual-trace log likelihoods with respect to $\boldsymbol{\theta}$. We refer to Equation 3 as the adjusted maximum likelihood objective. We see that the first term fits the GP to the outcome data, and the second term acts as an adjustment to account for dependencies between future outcomes and the timing and types of actions that were observed in the training data.

**Connection to target counterfactual.** By maximizing Equation 3, we obtain a statistical model of the observational traces $\mathcal{D}$. In general, the statistical model may not recover the target counterfactual model (Equation 1). To connect the CGP to Equation 1, we describe two additional assumptions. The first assumption is an alternative to Assumption 2.

**Assumption 3** (Continuous-Time NUC). *For all times $t$ and all histories $\mathcal{H}_{t^-}$, the densities $\lambda^*(t)$, $p^*(z_y, z_a \mid t)$, and $p^*(a \mid y, t, z_a)$ do not depend on $Y_s[\mathbf{a}]$ for all times $s > t$ and all actions $\mathbf{a}$.*

The key implication of this assumption is that the policy used to choose actions in the observational data did not depend on any unobserved information that is predictive of the future potential outcomes.

**Assumption 4** (Non-Informative Measurement Times). *For all times $t$ and any history $\mathcal{H}_{t^-}$, the following holds: $p^*(y \mid t, z_y = 1) \, \mathrm{d}y = P(Y_t \in \mathrm{d}y \mid \mathcal{H}_{t^-})$.*

Under Assumptions 1, 3, and 4, we can show that Equation 1 is equivalent to the GP used to model $p^*(y \mid t, z_y = 1)$. In the interest of space, the argument for this equivalence is in Section A of the supplement. Note that these assumptions are not statistically testable (see e.g., Pearl 2009).

## 3 Experiments

We demonstrate the CGP on two decision-support tasks. First, we show that the CGP can make reliable risk predictions that are insensitive to the action policy in the training data. Classical supervised learning algorithms, however, are dependent on the action policy and this can make them unreliable decision-support tools. Second, we show how the CGP can be used to compare counterfactuals and ask "what if?" questions for individualized treatment planning by learning the effects of dialysis on creatinine levels using real data from an intensive care unit (ICU).

|  | Regime $A$ | | Regime $B$ | | Regime $C$ | |
|---|---|---|---|---|---|---|
|  | Baseline GP | CGP | Baseline GP | CGP | Baseline GP | CGP |
| Risk Score $\Delta$ from $A$ | 0.000 | 0.000 | 0.083 | 0.001 | 0.162 | 0.128 |
| Kendall's $\tau$ from $A$ | 1.000 | 1.000 | 0.857 | 0.998 | 0.640 | 0.562 |
| AUC | 0.853 | 0.872 | 0.832 | 0.872 | 0.806 | 0.829 |

Table 1: Results measuring reliability for simulated data experiments. See Section 3.1 for details.

## 3.1 Reliable Risk Prediction with CGPs

We first show how the CGP can be used for reliable risk prediction, where the objective is to predict the likelihood of an adverse event so that we can intervene to prevent it from happening. In this section, we use simulated data so that we can evaluate using the true risk on test data. For concreteness, we frame our experiment within a healthcare setting, but the ideas can be more broadly applied. Suppose that a clinician records a real-valued measurement over time that reflects an individual's health, which we call a *severity marker*. We consider the individual to *not* be at risk if the severity marker is unlikely to fall below a particular threshold in the future without intervention. As discussed by Caruana et al. [2015], modeling this notion of risk can help doctors decide when an individual can safely be sent home without aggressive treatment.

We simulate the value of a severity marker recorded over a period of 24 hours in the hospital; high values indicate that the patient is healthy. A natural approach to predicting risk at time $t$ is to model the conditional distribution of the severity marker's future trajectory given the history up until time $t$; i.e. $P(\{Y_s : s > t\} \mid \mathcal{H}_t)$. We use this as our baseline. As an alternative, we use the CGP to explicitly model the counterfactual "What if we do not treat this patient?"; i.e. $P(\{Y_s[\varnothing] : s > t\} \mid \mathcal{H}_t)$. For all experiments, we consider a single decision time $t = 12$hrs. To quantify risk, we use the negative of each model's predicted value at the end of 24 hours, normalized to lie in $[0, 1]$.

**Data.** We simulate training and test data from three regimes. In regimes $A$ and $B$, we simulate severity marker trajectories that are treated by policies $\pi_A$ and $\pi_B$ respectively, which are both unknown to the baseline model and CGP at train time. Both $\pi_A$ and $\pi_B$ are designed to satisfy Assumptions 1, 3, and 4. In regime $C$, we use a policy that *does not* satisfy these assumptions. This regime will demonstrate the importance of verifying whether the assumptions hold when applying the CGP. We train both the baseline model and CGP on data simulated from all three regimes. We test all models on a common set of trajectories treated up until $t = 12$hrs with policy $\pi_A$ and report how risk predictions vary as a function of action policy in the training data.

**Simulator.** For each patient, we randomly sample outcome measurement times from a homogeneous Poisson process with with constant intensity $\lambda$ over the 24 hour period. Given the measurement times, outcomes are sampled from a mixture of three GPs. The covariance function is shared between all classes, and is defined using a Matérn $3/2$ kernel (variance $0.2^2$, lengthscale $8.0$) and independent Gaussian noise (scale $0.1$) added to each observation. Each class has a distinct mean function parameterized using a 5-dimensional, order-3 B-spline. The first class has a declining mean trajectory, the second has a trajectory that declines then stabilizes, and the third has a stable trajectory.[2] All classes are equally likely *a priori*. At each measurement time, the treatment policy $\pi$ determines a probability $p$ of treatment administration (we use only a single treatment type). The treatments increase the severity marker by a constant amount for 2 hours. If two or more actions occur within 2 hours of one another, the effects do not add up (i.e. it is as though only one treatment is active). Additional details about the simulator and policies can be found in the supplement.

**Model.** For both the baseline GP and CGP, we use a mixture of three GPs (as was used to simulate the data). We assume that the mean function coefficients, the covariance parameters, and the treatment effect size are unknown and must be learned. We emphasize that both the baseline GP and CGP have identical forms, but are trained using different objectives; the baseline marginalizes over future actions, inducing a dependence on the treatment policy in the training data, while the CGP explicitly controls for them while learning. For both the baseline model and CGP, we analytically sum over the mixture component likelihoods to obtain a closed form expression for the likelihood, which we optimize using BFGS [Nocedal and Wright, 2006].[3] Predictions for both models are made using the posterior predictive mean given data and interventions up until 12 hours.

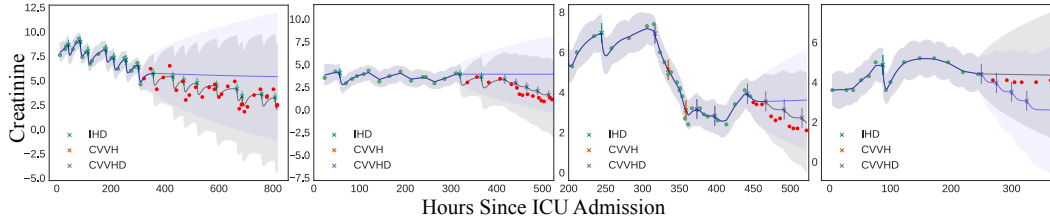

Figure 2: Example factual (grey) and counterfactual (blue) predictions on real ICU data using the CGP.

**Results.** We find that the baseline GP's risk scores fluctuate across regimes $A$, $B$, and $C$. The CGP is stable across regimes $A$ and $B$, but unstable in regime $C$, where our assumptions are violated. In Table 1, the first row shows the average difference in risk scores (which take values in $[0, 1]$) produced by the models trained in each regime and produced by the models trained in regime $A$. In row 1, column $B$ we see that the baseline GP's risk scores differ for the same person on average by around eight points ($\Delta = 0.083$). From the perspective of a decision-maker, this behavior could make the system appear less reliable. Intuitively, the risk for a given patient should not depend on the policy used to determine treatments in retrospective data. On the other hand, the CGP's scores change very little when trained on different regimes ($\Delta = 0.001$), as long as Assumptions 1, 3, and 4 are satisfied.

A cynical reader might ask: even if the risk scores are unstable, perhaps it has no consequences on the downstream decision-making task? In the second row of Table 1, we report Kendall's $\tau$ computed between each regime and regime $A$ using the risk scores to rank the patient's in the test data according to severity (i.e. scores closer to 1 are more severe). In the third row, we report the AUC for both models trained in each regime on the common test set. We label a patient as "at risk" if the last marker value in the untreated trajectory is below zero, and "not at risk" otherwise. In row 2, column $B$ we see that the CGP has a high rank correlation ($\tau = 0.998$) between the two regimes where the policies satisfy our key assumptions. The baseline GP model trained on regime $B$, however, has a lower rank correlation of $\tau = 0.857$ with the risk scores produced by the same model trained on regime $A$. Similarly, in row three, columns $A$ and $B$, we see that the CGP's AUC is unchanged (AUC $= 0.872$). The baseline GP, however, is unstable and creates a risk score with poorer discrimination in regime $B$ (AUC $= 0.832$) than in regime $A$ (AUC $= 0.853$). Although we illustrate stability of the CGP compared to the baseline GP using two regimes, this property is not specific to the particular choice of policies used in regimes $A$ and $B$; the issue persists as we generate different training data by varying the distribution over the action choices.

Finally, the results in column $C$ highlight the importance of Assumptions 1, 3, and 4. The policy $\pi_C$ *does not* satisfy these assumptions, and we see that the risk scores for the CGP are different when fit in regime $C$ than when fit in regime $A$ ($\Delta = 0.128$). Similarly, in row 2 the CGP's rank correlation degrades ($\tau = 0.562$), and in row 3 the AUC decreases to $0.829$. Note that the baseline GP continues to be unstable when fit in regime $C$.

**Conclusions.** These results have important implications for the practice of building predictive models for decision support. Classical supervised learning algorithms can be unreliable due to an implicit dependence on the action policy in the training data, which is usually different from the assumed action policy at test time (e.g. what will happen if we do not treat?). Note that this issue is not resolved by training only on individuals who are not treated because selection bias creates a mismatch between our train and test distributions. From a broader perspective, supervised learning can be unreliable because it captures features of the training distribution that may change (e.g. relationships caused by the action policy). Although we have used a counterfactual model to account for and remove these unstable relationships, there may be other approaches that achieve the same effect (e.g., Dyagilev and Saria 2016). Recent related work by Gong et al. [2016] on covariate shift aims to learn only the components of the source distribution that will generalize to the target distribution. As predictive models are becoming more widely used in domains like healthcare where safety is critical (e.g. Li-wei et al. 2015, Schulam and Saria 2015, Alaa et al. 2016, Wiens et al. 2016, Cheng et al. 2017), the framework proposed here is increasingly pertinent.

### 3.2 "What if?" Reasoning for Individualized Treatment Planning

To demonstrate how the CGP can be used for individualized treatment planning, we extract observational creatinine traces from the publicly available MIMIC-II database [Saeed et al., 2011].

Creatinine is a compound produced as a by-product of the chemical reaction in the body that breaks down creatine to fuel muscles. Healthy kidneys normally filter creatinine out of the body, which can otherwise be toxic in large concentrations. During kidney failure, however, creatinine levels rise and the compound must be extracted using a medical procedure called dialysis.

We extract patients in the database who tested positive for abnormal creatinine levels, which is a sign of kidney failure. We also extract the times at which three different types of dialysis were given to each individual: intermittent hemodialysis (IHD), continuous veno-venous hemofiltration (CVVH), and continuous veno-venous hemodialysis (CVVHD). The data set includes a total of 428 individuals, with an average of 34 ($\pm$12) creatinine observations each. We shuffle the data and use 300 traces for training, 50 for validation and model selection, and 78 for testing.

**Model.** We parameterize the outcome model of the CGP using a mixture of GPs. We always condition on the initial creatinine measurement and model the deviation from that initial value. The mean for each class is zero (i.e. we assume there is no deviation from the initial value on average). We parameterize the covariance function using the sum of two non-stationary kernel functions. Let $\phi : t \to [1, t, t^2]^\top \in \mathbb{R}^3$ denote the quadratic polynomial basis, then the first kernel is $k_1(t_1, t_2) = \phi^\top(t_1)\Sigma\phi(t_2)$, where $\Sigma \in \mathbb{R}^{3 \times 3}$ is a positive-definite symmetric matrix parameterizing the kernel. The second kernel is the covariance function of the integrated Ornstein-Uhlenbeck (IOU) process (see e.g., Taylor et al. 1994), which is parameterized by two scalars $\alpha$ and $\nu$ and defined as

$$k_{\text{IOU}}(t_1, t_2) = \tfrac{\nu^2}{2\alpha^3} \left( 2\alpha \min(t_1, t_2) + e^{-\alpha t_1} + e^{-\alpha t_2} - 1 - e^{-\alpha|t_1 - t_2|} \right).$$

The IOU covariance corresponds to the random trajectory of a particle whose velocity drifts according to an OU process. We assume that each creatinine measurement is observed with independent Gaussian noise with scale $\sigma$. Each class in the mixture has a unique set of covariance parameters. To model the treatment effects in the outcome model, we define a short-term function and long-term response function. If an action is taken at time $t_0$, the outcome $\delta = t - t_0$ hours later will be additively affected by the response function $g(\delta; h_1, a, b, h_2, r) = g_s(\delta; h_1, a, b) + g_\ell(\delta; h_2, r)$, where $h_1, h_2 \in \mathbb{R}$ and $a, b, r \in \mathbb{R}^+$. The short-term and long-term response functions are defined as $g_s(\delta; h_1, a, b) = \frac{h_1 a}{a-b} \left( e^{-b \cdot t} - e^{-a \cdot t} \right)$, and $g_\ell(\delta : h_2, r) = h_2 \cdot (1.0 - e^{-r \cdot t})$. The two response functions are included in the mean function of the GP, and each class in the mixture has a unique set of response function parameters. We assume that Assumptions 1, 3, and 4 hold, and that the event and action models have separate parameters, so can remain unspecified when estimating the outcome model. We fit the CGP outcome model using Equation 3, and select the number of classes in the mixture using fit on the validation data (we choose three components).

**Results.** Figure 2 demonstrates how the CGP can be used to do "what if?" reasoning for treatment planning. Each panel in the figure shows data for an individual drawn from the test set. The green points show measurements on which we condition to obtain a posterior distribution over mixture class membership and the individual's latent trajectory under each class. The red points are unobserved, future measurements. In grey, we show predictions under the *factual* sequence of actions extracted from the MIMIC-II database. Treatment times are shown using vertical bars marked with an "x" (color indicates which type of treatment was given). In blue, we show the CGP's *counterfactual* predictions under an alternative sequence of actions. The posterior predictive trajectory is shown for the MAP mixture class (mean is shown by a solid grey/blue line, 95% credible intervals are shaded).

We qualitatively discuss the CGP's counterfactual predictions, but cannot quantitatively evaluate them without prospective experimental data from the ICU. We can, however, measure fit on the factual data and compare to baselines to evaluate our modeling decisions. Our CGP's outcome model allows for heterogeneity in the covariance parameters and the response functions. We compare this choice to two alternatives. The first is a mixture of three GPs that *does not* model treatment effects. The second is a single GP that *does* model treatment effects. Over a 24-hour horizon, the CGP's mean absolute error (MAE) is 0.39 (95% CI: 0.38-0.40),[4] and for predictions between 24 and 48 hours in the future the MAE is 0.62 (95% CI: 0.60-0.64). The pairwise mean difference between the first baseline's absolute errors and the CGP's is 0.07 (0.06, 0.08) for 24 hours, and 0.09 (0.08, 0.10) for 24-48 hours. The mean difference between the second baseline's absolute errors and the CGP's is 0.04 (0.04, 0.05) for 24 hours and 0.03 (0.02, 0.04) for 24-48 hours. The improvements over the baselines suggest that modeling treatments and heterogeneity with a mixture of GPs for the outcome model are useful for this problem.

Figure 2 shows factual and counterfactual predictions made by the CGP. In the first (left-most) panel, the patient is factually administered IHD about once a day, and is responsive to the treatment (creatinine steadily improves). We query the CGP to estimate how the individual *would have* responded had the IHD treatment been stopped early. The model reasonably predicts that we would have seen no further improvement in creatinine. The second panel shows a similar case. In the third panel, an individual with erratic creatinine levels receives CVVHD for the last 100 hours and is responsive to the treatment. As before, the CGP counterfactually predicts that she would not have improved had CVVHD not been given. Interestingly, panel four shows the opposite situation: the individual did not receive treatment and did not improve for the last 100 hours, but the CGP counterfactually predicts an improvement in creatinine as in panel 3 under daily CVVHD.

## 4 Discussion

Classical supervised learning algorithms can lead to unreliable and, in some cases, dangerous decision-support tools. As a safer alternative, this paper advocates for using potential outcomes [Neyman, 1923, 1990, Rubin, 1978] and *counterfactual learning objectives* (like the one in Equation 3). We introduced the Counterfactual Gaussian Process (CGP) as a decision-support tool for scenarios where outcomes are measured and actions are taken at irregular, discrete points in continuous-time. The CGP builds on previous ideas in continuous-time causal inference (e.g. Robins 1997, Arjas and Parner 2004, Lok 2008), but is unique in that it can predict the full counterfactual *trajectory* of a time-dependent outcome. We designed an adjusted maximum likelihood algorithm for learning the CGP from *observational traces* by modeling them using a marked point process (MPP), and described three structural assumptions that are sufficient to show that the algorithm correctly recovers the CGP.

We empirically demonstrated the CGP on two decision-support tasks. First, we showed that the CGP can be used to make reliable risk predictions that are insensitive to the action policies used in the training data. This is critical because an action policy can cause a predictive model fit using classical supervised learning to capture relationships between the features and outcome (risk) that lead to poor downstream decisions and that are difficult to diagnose. In the second set of experiments, we showed how the CGP can be used to compare counterfactuals and answer "what if?" questions, which could offer decision-makers a powerful new tool for individualized treatment planning. We demonstrated this capability by learning the effects of dialysis on creatinine trajectories using real ICU data and predicting counterfactual progressions under alternative dialysis treatment plans.

These results suggest a number of new questions and directions for future work. First, the validity of the CGP is conditioned upon a set of assumptions (this is true for all counterfactual models). In general, these assumptions are not testable. The reliability of approaches using counterfactual models therefore critically depends on the plausibility of those assumptions in light of domain knowledge. Formal procedures, such as sensitivity analyses (e.g., Robins et al. 2000, Scharfstein et al. 2014), that can identify when causal assumptions conflict with a data set will help to make these methods more easily applied in practice. In addition, there may be other sets of structural assumptions beyond those presented that allow us to learn counterfactual GPs from non-experimental data. For instance, the back door and front door criteria are two separate sets of structural assumptions discussed by Pearl [2009] in the context of estimating parameters of causal Bayesian networks from observational data.

More broadly, this work has implications for recent pushes to introduce safety, accountability, and transparency into machine learning systems. We have shown that learning algorithms sensitive to certain factors in the training data (the action policy, in this case) can make a system less reliable. In this paper, we used the potential outcomes framework and counterfactuals to characterize and account for such factors, but there may be other ways to do this that depend on fewer or more realistic assumptions (e.g., Dyagilev and Saria 2016). Moreover, removing these nuisance factors is complementary to other system design goals such as interpretability (e.g., Ribeiro et al. 2016).

**Acknowledgements**

We thank the anonymous reviewers for their insightful feedback. This work was supported by generous funding from DARPA YFA #D17AP00014 and NSF SCH #1418590. PS was also supported by an NSF Graduate Research Fellowship. We thank Katie Henry and Andong Zhan for help with the ICU data set. We also thank Miguel Hernán for pointing us to earlier work by James Robins on treatment-confounder feedback.

## Footnotes

[1] $y_{ij}$ and $a_{ij}$ may be the null variable $\varnothing$ to allow for the possibility that an action is taken but no outcome is observed and vice versa. $[0, \tau]$ denotes a fixed period of time over which the trajectories are observed.

[2]The exact B-spline coefficients can be found in the simulation code included in the supplement.

[3]Additional details can be found in the supplement.

[4]95% confidence intervals computed using the pivotal bootstrap are shown in parentheses

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
