[Supplementary Material]

# Reliable Decision Support using Counterfactual Models

**Peter Schulam**
Department of Computer Science
Johns Hopkins University
Baltimore, MD 21211
pschulam@cs.jhu.edu

**Suchi Saria**
Department of Computer Science
Johns Hopkins University
Baltimore, MD 21211
ssaria@cs.jhu.edu

## A  Equivalence of MPP Outcome Model and Counterfactual Model

At a given time $t$, we want to make predictions about the potential outcomes that we will measure at a set of future query times $\mathbf{q} = [s_1, \ldots, s_m]$ given a specified future sequence of actions $\mathbf{a}$. This can be written formally as

$$P(\{Y_s[\mathbf{a}] : s \in \mathbf{q}\} \mid \mathcal{H}_t) \tag{1}$$

Without loss of generality, we can use the chain rule to factor this joint distribution over the potential outcomes. We choose a factorization in time order; that is, a potential outcome is conditioned on all potential outcomes at earlier times. We now describe a sequence of steps that we can apply to each factor in the product.

$$P(\{Y_s[\mathbf{a}] : s \in \mathbf{q}\} \mid \mathcal{H}_t) = \prod_{i=1}^{m} P(Y_{s_i}[\mathbf{a}] \mid \{Y_s[\mathbf{a}] : s \in \mathbf{q}, s < s_i\}, \mathcal{H}_t). \tag{2}$$

Using Assumption 3, we can introduce random variables for marked points that have the same timing and actions as the proposed sequence of actions without changing the probability. Recall our assumption that actions can only affect future values of the outcome, so we only need to introduce marked points for actions taken at earlier times. Formally, we introduce the set of marked points for the potential outcome at each time $s_i$

$$\mathbf{A}_i = \{(t', \varnothing, a, 0, 1) : (t', a) \in \mathbf{a}, t' < s_i\}. \tag{3}$$

We can then write

$$P(Y_{s_i}[\mathbf{a}] \mid \{Y_s[\mathbf{a}] : s \in \mathbf{q}, s < s_i\}, \mathcal{H}_t) = P(Y_{s_i}[\mathbf{a}] \mid \mathbf{A}_i, \{Y_s[\mathbf{a}] : s \in \mathbf{q}, s < s_i\}, \mathcal{H}_t). \tag{4}$$

To show that $P(Y[a] \mid A = a, X = x) = P(Y[a] \mid X = x)$ in Section 2, we use Assumption 2 to remove the random variable $A$ from the conditioning information without changing the probability statement. We reverse that logic here by adding $\mathbf{A}_i$.

Now, under Assumption 1, after conditioning on $\mathbf{A}_i$, we can replace the potential outcome $Y_{s_i}[\mathbf{a}]$ with $Y_{s_i}$. We therefore have

$$P(Y_{s_i}[\mathbf{a}] \mid \mathbf{A}_i, \{Y_s[\mathbf{a}] : s \in \mathbf{q}, s < s_i\}, \mathcal{H}_t) = P(Y_{s_i} \mid \mathbf{A}_i, \{Y_s[\mathbf{a}] : s \in \mathbf{q}, s < s_i\}, \mathcal{H}_t). \tag{5}$$

Similarly, because the set of proposed actions affecting the outcome at time $s_i$ contain all actions that affect the outcome at earlier times $s < s_i$, we can invoke Assumption 1 again and replace all potential outcomes at earlier times with the value of the observed process at that time.

$$P(Y_{s_i} \mid \mathbf{A}_i, \{Y_s[\mathbf{a}] : s \in \mathbf{q}, s < s_i\}, \mathcal{H}_t) = P(Y_{s_i} \mid \mathbf{A}_i, \{Y_s : s \in \mathbf{q}, s < s_i\}, \mathcal{H}_t).$$

Figure 1: The causal Bayesian network for the counterfactual GP.

Next, Assumption 4 posits that the outcome model $p^*(y \mid t', z_y = 1)$ is the density of $P(Y_{t'} \mid \mathcal{H}_t)$, which implies that the mark $(t', y, \varnothing, 1, 0)$ is equivalent to the event $(Y_{t'} \in \mathrm{d}y)$. Therefore, for each $s_i$ define

$$\mathbf{O}_i = \{(s, Y_s, \varnothing, 1, 0) : s \in \mathbf{q}, s < s_i\}. \tag{6}$$

Using this definition, we can write

$$P(Y_{s_i} \mid \mathbf{A}_i, \{Y_s : s \in \mathbf{q}, s < s_i\}, \mathcal{H}_t) = (Y_{s_i} \mid \mathbf{A}_i, \mathbf{O}_i, \mathcal{H}_t).$$

The set of information $(\mathbf{A}_i, \mathbf{O}_i, \mathcal{H}_t)$ is a valid history of the marked point process $\mathcal{H}_{s_i}^-$ up to but not including time $s_i$. We can therefore replace all information after the conditioning bar in each factor of Equation 2 with $\mathcal{H}_{s_i^-}$.

$$P(Y_{s_i} \mid \mathbf{A}_i, \mathbf{O}_i, \mathcal{H}_t) = P(Y_{s_i} \mid \mathcal{H}_{s_i}^-). \tag{7}$$

Finally, by applying Assumption 4 again, we have

$$P(Y_{s_i} \in \mathrm{d}y \mid \mathcal{H}_{s_i}^-) = p^*(y \mid s_i, z_y = 1)\, \mathrm{d}y. \tag{8}$$

The potential outcome query can therefore be answered using the outcome model, which we can estimate from data.

## B    Causal Bayesian Network

We can also characterize our key assumptions using causal Bayesian networks [Pearl, 2009]. Let $\{(t_j, z_{y,j}, z_{a,j}, y_j, a_j)\}_{j \geq 1}$ be a countable sequence of tuples of variables (a marked point process can be characterized as a countable sequence of points and marks). Recall that $t_j$ is an event time, $z_{y,j}$ is a binary random variable indicating whether an outcome is measured, $z_{a,j}$ is a binary random variable indicating whether an action is taken, $y_j \in \mathcal{R} \cup \{\varnothing\}$ is an outcome measurement, and $a_j \in \mathcal{C} \cup \{\varnothing\}$ is an action (the last two variables are $\varnothing$ when the respective indicator is 0).

We define the directed acyclic graph $\mathcal{G}$ with nodes $\mathcal{V} \triangleq \cup_{j \geq 1} \{t_j, z_{y,j}, z_{a,j}, y_j, a_j\}$ and edge set $\mathcal{E}$ to be the causal Bayesian network for the counterfactual GP. For any variables $v_1 \in \{t_j, z_{y,j}, z_{a,j}, y_j, a_j\}$ and $v_2 \in \{t_k, z_{y,k}, z_{a,k}, y_k, a_k\}$, the edge $(v_1 \to v_k) \in \mathcal{E}$ if $j < k$ or if $j = k$ and $v_1$ is a parent of $v_2$ in the right-most plate of Figure 1. We allow the variables $\{(t_j, z_{y,j}, z_{a,j}, a_j)\}_{j=1}^{\infty}$ to depend on a common unobserved parent $u_1$, and the outcomes $\{y_j\}_{j=1}^{\infty}$

to depend on a common unobserved parent $u_2$. The DAG in Figure 1 sketches the causal Bayesian network. For any index $j$, we show the edges present between all variables at times $k < j$.

We now formulate our causal query, and show that it is identified using observational traces sampled from the distribution implied by the causal Bayesian network. For any time $t \in [0, \tau]$, our goal is to predict the values of future outcomes under a hypothetical sequence of future actions given the history up until time $t$. Define $\mathcal{H}_t = \cup_{j:t_j<t}\{t_j, z_{y,j}, z_{a,j}, y_j, a_j\}$ to be the sequence of $n$ actions taken and outcomes measured prior to time $t$, and define $\mathcal{F}_t$ to be a sequence of $m$ tuples corresponding to future actions and measurements. The variables in $\mathcal{H}_t \cup \mathcal{F}_t$ are connected using the edge set definition described above. Let $\boldsymbol{t}$ denote the $m$ future time points, $\boldsymbol{z}_y$ the future measurement indicators, $\boldsymbol{z}_a$ the future action indicators, $\boldsymbol{y}$ the future outcomes, and $\boldsymbol{a}$ the future actions. Our goal is to show that the following query is identified:

$$p(\boldsymbol{y} \mid \mathrm{do}(\boldsymbol{t}, \boldsymbol{z}_y, \boldsymbol{z}_a, \boldsymbol{a}), \mathcal{H}_t) = \prod_{j=1}^{m} p(y_j \mid \bar{\boldsymbol{y}}_{:j}, \mathrm{do}(\boldsymbol{t}, \boldsymbol{z}_y, \boldsymbol{z}_a, \boldsymbol{a}), \mathcal{H}_t), \qquad (9)$$

where $\bar{\boldsymbol{y}}_{:j}$ denotes the vector of future outcomes before the $j^{\text{th}}$. We will also use $\bar{\boldsymbol{y}}_{j:}$ to denote all outcomes measured after the $j^{\text{th}}$ (this notation will be used for the other variables as well). First, consider any factor in the expression above. We define the future and past intervened-on variables at time $t_j$ as

$$\boldsymbol{f}_j \triangleq \{a_j, \bar{\boldsymbol{t}}_{j:}, \bar{\boldsymbol{z}}_{y,j:}, \bar{\boldsymbol{z}}_{a,j:}, \bar{\boldsymbol{a}}_{j:}\} \qquad (10)$$
$$\boldsymbol{p}_j \triangleq \{\bar{\boldsymbol{t}}_{:j}, \bar{\boldsymbol{z}}_{y,:j}, \bar{\boldsymbol{z}}_{a,:j}, \bar{\boldsymbol{a}}_{:j}, t_j, z_{y,j}, z_{a,j}\}. \qquad (11)$$

Using these shorthand definitions, we first prove the following equivalence

$$p(y_j \mid \bar{\boldsymbol{y}}_{:j}, \mathrm{do}(\boldsymbol{p}_j), \mathrm{do}(\boldsymbol{f}_j), \mathcal{H}_t) = p(y_j \mid \bar{\boldsymbol{y}}_{:j}, \mathrm{do}(\boldsymbol{p}_j), \mathcal{H}_t). \qquad (12)$$

Intuitively, we are showing that actions taken after $y_j$ is measured do not affect its value. To justify the equality, we use "Rule 3" from Pearl's do-calculus (see Chapter 3 in Pearl 2009). We must show that $y_j$ is d-separated from $\boldsymbol{f}_j$ in the mutilated DAG where all incoming edges to nodes in $\boldsymbol{p}_j$ and $\boldsymbol{f}_j$ have been removed. To show d-separation, let $v \in \boldsymbol{f}_j \setminus \{a_j\}$ be some future intervened-on variable at time step $k > j$. Since all incoming edges have been removed, all paths starting at $v$ must be outgoing. Outgoing edges for $v$ in the original DAG either point to an outcome $y_\ell$ for $\ell \geq k$ or some other intervened-on variable $v' \in \boldsymbol{f}_j \setminus \{a_j, v\}$. The latter are removed in the mutilated graph, so the only edges outgoing from $v$ must point to an outcome $y_\ell$ for $\ell \geq k$. This implies that all paths starting at $v$ must begin with an edge $v \to y_\ell$ for some $\ell \geq k$. Because $y_\ell$ is unobserved, the only unblocked paths must then follow an outgoing edge (otherwise it would be a collider). All outgoing edges from variables $y_\ell$ for $\ell \geq k$ can only point to outcomes $y_{\ell'}$ for $\ell' > \ell$, which in turn must point to $y_{\ell''}$ for $\ell'' > \ell'$, and so on. Therefore, any path starting from $v$ must pass through outcomes $y$ at strictly increasing times. Eventually, we will reach the final outcome, where there are no outgoing edges, ending the path. We can conclude that no paths starting at $v$ can reach $y_j$. A similar argument shows that no path starting from $a_j$ can reach $y_j$.

Next, we use "Rule 2" from the do-calculus to prove that

$$p(y_j \mid \bar{\boldsymbol{y}}_{:j}, \mathrm{do}(\boldsymbol{p}_j), \mathcal{H}_t) = p(y_j \mid \bar{\boldsymbol{y}}_{:j}, \boldsymbol{p}_j, \mathcal{H}_t). \qquad (13)$$

This requires showing that $y_j$ is d-separated from $\boldsymbol{p}_j$ in the mutilated graph where all outgoing edges from $v \in \boldsymbol{p}_j$ have been removed. For any $v \in \boldsymbol{p}_j$, there are two types of incoming edges. The first are edges originating from observed direct parents of $v$, and the second is the edge originating from the unobserved variable $u_1$. Any path from $v$ to $y_j$ must start with one of these edge types, and therefore all that start with an edge to an observed parent of $v$ will be blocked, and any unblocked path must start by going through $u_1$. Now, $u_1$ has no parents and any path must then have a second edge from $u_1$ to one of its children, which are all times $t_k$, indicators $z_{y,k}$ or $z_{a,k}$, and actions $a_k$. We will analyze these possibilities using two cases. First, the second edge could go from $u_1$ to a time $t_k$ where $k \leq j$, indicator $z_{y,k}$ or $z_{a,k}$ where $k \leq j$, or to an action $a_k$ where $k < j$. The only possible next step is to go through an incoming edge where the origin is not $u_1$; all such edges will be blocked, and so cannot reach $y_j$. In the second case, an edge could go from $u_1$ to a time or indicator at step $k > j$, or an action at step $k \geq j$. These variables are unobserved, and so the only valid next step is to follow an outgoing edge. Subsequent steps must all also follow outgoing edges by the same logic,

and so the path can never return to $y_j$. We therefore can conclude that there are no paths from $v \in \boldsymbol{p}_j$ to $y_j$ in the mutilated graph, so the equality holds. Together, the two inequalities show

$$p(\boldsymbol{y} \mid \mathrm{do}(\boldsymbol{t}, \boldsymbol{z}_y, \boldsymbol{z}_a, \boldsymbol{a}), \mathcal{H}_t) = \prod_{j=1}^{m} p(y_j \mid \bar{\boldsymbol{y}}_{:j}, \boldsymbol{p}_j, \mathcal{H}_t). \tag{14}$$

This shows that the structural dependencies encoded in the graph shown in Figure 1 can be used in place of Assumption 3. In addition, we no longer need Assumption 1 (consistency), which highlights an interesting difference between the potential outcomes and causal Bayesian network frameworks. In Pearl's causal DAGs, consistency is in fact a theorem derived from the axioms of the framework, whereas it is assumed in the potential outcomes framework. This is shown in Corollary 7.3.2 in Pearl [2009], which follows from the Composition axiom and the definition of a "null" intervention. Intuitively, the fact that consistency is a theorem in Pearl's framework reflects the assumption that the parent-child relationships in the DAG are sufficiently stable, autonomous, or "local" [Pearl, 2009]. See Section 7.2.4 in Pearl [2009] for further information. Finally, Assumption 4 remains unchanged and simply allows us to treat measured outcomes $y_j$ as unbiased samples of the process $Y_{t_j}$.

## C Simulation and Policy Details

For each patient, we randomly sample outcome measurement times from a homogeneous Poisson process with with constant intensity $\lambda$ over the 24 hour period. Given the measurement times, outcomes are sampled from a mixture of three GPs. The covariance function is shared between all classes, and is defined using a Matérn $3/2$ kernel (variance $0.2^2$, lengthscale $8.0$) and independent Gaussian noise (scale $0.1$) added to each observation. Each class has a distinct mean function parameterized using a 5-dimensional, order-3 B-spline. The first class has a declining mean trajectory, the second has a trajectory that declines then stabilizes, and the third has a stable trajectory.[1] All classes are equally likely *a priori*. At each measurement time, the treatment policy $\pi$ determines a probability $p$ of treatment administration (we use only a single treatment type). The treatments increase the severity marker by a constant amount for 2 hours. If two or more actions occur within 2 hours of one another, the effects do not add up (i.e. it is as though only one treatment is active). Additional details about the simulator and policies can be found in the supplement.

Policies $\pi_A$ and $\pi_B$ determine a probability of treatment at each outcome measurement time. They each use the average of the observed outcomes over the previous two hours, which we denote using $\hat{y}_{(t-2):t}$, as a feature, which is then multiplied by a weight $w_A = -0.5$ ($w_B = 0.5$ for regime B) and passed through the inverse logit to determine a probabilty. The policy $\pi_C$ for regime $C$ depends on the patient's latent class. The probability of treatment at any time $t$ is $p = \alpha_z \sigma(w_A \cdot \hat{y}_{(t-2):t})$, where $\alpha_z \in (0,1)$ is a weight that depends on the latent class $z$. We set $\alpha_1 = 0.2$, $\alpha_2 = 0.9$, and $\alpha_3 = 0.5$.

## D Mixture Estimation Details

For both the simulated and real data experiments, we analytically sum over the component-specific densities to obtain an explicit mixture density involving no latent variables. We then estimate the parameters using maximum likelihood. The likelihood surface is highly non-convex. To account for this, we used different parameter initialization strategies for the simulated and real data.

On the simulated data experiments, the mixture components for both the CGP and baseline GP are primarily distinguished by the mean functions. We initialize the mean parameters for both the baseline GP and CGP by first fitting a linear mixed model with B-spline bases using the EM algorithm, computing MAP estimates of trace-specific coefficients, clustering the coefficients, and initializing with the cluster centers.

On the real data, traces have similar mean behavior (trajectories drift around the initial creatinine value), but differed by length and amplitude of variations from the mean. We therefore centered each trace around its initial creatinine measurement (which we condition on), and use a mean function that includes only the short-term and long-term response functions. For each mixture, the response function parameters are initialized randomly: parameters $a$, $b$, and $r$ are initialized using

a LogNormal(mean $= 0.0$, std $= 0.1$); heights $h_1$ and $h_2$ are initialized using a Normal(mean $=$ 0.0, std $= 0.1$). For each mixture, $\Sigma$ (L300) is initialized to the identity matrix; $\alpha$ and $\nu$ are drawn from a LogNormal(mean $= 0.0$, std $= 0.1$).

## Footnotes

[1]The exact B-spline coefficients can be found in the simulation code included in the supplement.

## References

J. Pearl. *Causality: models, reasoning and inference*. Cambridge University Press, 2009.