[Reviews · NeurIPS 2017]

Reviewer 1



Comments: - formula (3). There are misprints in the formula. E.g. in the second term other arguments for lambda^*(t) should be used; in the third term lambda depends on theta, however theta are parameters of the CGP process and lambda in the second term does not depend on theta. Then is this a misprint? - Figure 2. The figure is pale, names of axes are too small - page 7, section "Model". 1) How did the authors combine these two covariance functions? 2) Used covariance functions depend only on time. How did the authors model depedence of outcome distribution on a history, actions, etc.? 3) In this section the authors introduced response functions for the first time. Why do they need them? How are they related with model (2)? - page 8. Discussion section. I would say that the idea to combine MPP and GP is not new in the sense that there already exist some attempts to use similar models in practice, see e.g. https://stat.columbia.edu/~cunningham/pdf/CunninghamAISTATS2012.pdf Conclusions: - the topic of the paper is important - the paper is very well written - a new applied model is proposed for a particular class of applications - although the idea is very simple, experimental results clearly show efficiency of the developed method - thus, this paper can be interesting for a corresponding ML sub-community

Reviewer 2



Thanks to the authors for a very interesting paper. The main contribution that I see is the proposal of a counterfactual GP model, which can take into account discrete actions, events, and outcomes, and as the authors argue allows for a much better model of discrete event data, e.g. electronic medical records, by explicitly accounting for future actions. A minor point: the citation for Neyman shouldn't be to 1990, as the paper is from 1923. Yes, the accessible translated version is from 1990, but the citation should include the fact that it's of an article from 1923, seeing as Neyman died in 1981. ;) I have some suggestions for strengthening the presentation and outlined some of what confused me. Hopefully addressing this will improve the paper. The framework adopted, of counterfactual predictions, is a very reasonable way to think about causal inference. I believe that the two main assumptions are standard in some of the literature, but, following Pearl, I do not think that they are the best way to think about what is going on. Overall, I would very much like to see a graphical model in Section 2. Regarding Assumption 1, I would like an account following or at least engaging with Pearl's critique of VanderWeele's claim about the consistency rule as an assumption rather than a theorem. Regarding Assumption 2, you cite Pearl so I am sure you are aware of his viewpoint, but I think it is very important to make clear what the no unmeasured confounders assumption is, and I believe that a graphical model is the best way to do this. It is worth pointing out as well that, following Pearl, we can go beyond the (often unrealistic) assumption of no unmeasured confounders (see e.g. the back-door and front-door criteria) as long as we're willing to state clearly what our structural assumptions are. Far from a criticism of how you've set things up, I believe that this viewpoint will mean you can cover more cases, as long as there is a way to setup the appropriate regression using a GP (which there should be!) Having I think understood Assumptions 1 and 2, it now seems like the claim is that we can estimate P(Y[a]) by marginalizing P(Y | A = a, X = x) with respect to X. But I'm having trouble connecting this with the claim that the CGP will do better than the RGP. The end of the "Model." section has the key insight, which I think you ought to expand a bit on and move earlier. It'd be helpful, I think, to write down the likelihood of the RGP so that it can be compared to Eq (3). It would also help to refer to Figure 1 (which is indeed best viewed in color, but could be made quite a bit more legible in grayscale!)---what would RGP do for Figure 1, for example? Now having read the experiments I'm confused again. In the "Experimental" scenario it is the case that after 12 hours there is a structural break, and all patients are assigned to a control condition, since no actions (treatments) are taken. But why are the RGP and CGP equivalent here? The Observational scenario makes sense, and the fact that the CGP performs well is reassuring, but why does this support your claim that the CGP is able to "learn a counterfactual predictive model"? Shouldn't you ask the CGP to make predictions under different treatment regimens---to take the example from Figure 1, shouldn't you ask the CGP to predict what will happen if Drug A is administered at hour 13, if Drug B is administered at hour 13, or if no drug is administered? It seems to me that the CGP is a better predictive model than the RGP, and thus does a better job predicting. I think it's true that this means that under your assumptions it will also do a better job on counterfactuals, but you should evaluate this. A question: how did you fit your models? I guess you maximized Equation 3, but there's some mixture modeling stuff going on here, and the model is highly non-convex. Were there issues with optimization? Did you need to do random restarts? How did you pick initial values? Where did the confidence intervals on parameters come from?

Reviewer 3



The paper presented a counterfactual model for time-continuous data building upon Gaussian processes. The literature review is extensive and thorough, and the experimental results are convincing. I have the following questions/comments: 1. To what extend one has to use the counterfactual arguments in order to get the final Bayesian model, which under assumptions 1 and 2 is sufficient for inference? 2. In (3), what is the model for $\lambda_\theta^*(s)$. Is it a parametric form? How is the integration done? 3. In all the examples, the observations are constrained to be non-negative. What is the likelihood model for the observation model that allows this to be done? Or is this restriction ignored and a Guasssian likelihood used? 4. How is the mixture of three GPs done? Is it via variational inference of MCMC? 5. In lines 329 and 330, it will be clearer if expressions for the two alternatives are given. Is the first of these the so called RGP?